# Severe Immune-Related Adverse Events in Patients Treated with Nivolumab for Metastatic Renal Cell Carcinoma Are Associated with *PDCD1* Polymorphism

**DOI:** 10.3390/genes13071204

**Published:** 2022-07-05

**Authors:** Mizuki Kobayashi, Kazuyuki Numakura, Shingo Hatakeyama, Yumina Muto, Yuya Sekine, Hajime Sasagawa, Soki Kashima, Ryohei Yamamoto, Atsushi Koizumi, Taketoshi Nara, Mitsuru Saito, Shintaro Narita, Chikara Ohyama, Tomonori Habuchi

**Affiliations:** 1Department of Urology, Akita University Graduate School of Medicine, Akita 010-8543, Japan; mkoba@med.akita-u.ac.jp (M.K.); yumina.muto.0601@gmail.com (Y.M.); ysekine.bs@gmail.com (Y.S.); sasahazi.820@gmail.com (H.S.); s4005534@gmail.com (S.K.); ryoheiyama815@yahoo.co.jp (R.Y.); koizu3atsu4@yahoo.co.jp (A.K.); taketonr@gipc.akita-u.ac.jp (T.N.); urosaito@gmail.com (M.S.); nari6202@gipc.akita-u.ac.jp (S.N.); thabuchi@gmail.com (T.H.); 2Department of Urology, Hirosaki University Graduate School of Medicine, Hirosaki 036-8203, Japan; shingorilla2@gmail.com (S.H.); coyama@hirosaki-u.ac.jp (C.O.)

**Keywords:** single nucleotide polymorphism, *PDCD1*, nivolumab, adverse event, metastatic renal cell carcinoma

## Abstract

Single nucleotide polymorphisms (SNPs) reportedly influence the effect of nivolumab in metastatic renal cell carcinoma (mRCC). This study aimed to evaluate the relationship between the clinical outcomes of patients with mRCC and SNPs in programmed cell death protein 1 (PD-1) protein-coding gene (*PDCD1*) and explore any potential correlation with patient prognosis and incidence of immune-related adverse events (irAEs). In total, 106 patients with mRCC, who were treated with nivolumab alone (*n* = 59) or nivolumab and ipilimumab (*n* = 47), were enrolled in the study. Three SNPs in the *PDCD1* gene, namely *PD-1.3*, *PD-1.5*, and *PD-1.6*, were assessed. Patients harboring the *PD-1.6 G* allele experienced more severe (odds ratio, 3.390; 95% confidence interval 1.517–7.756; *p* = 0.003) and multiple (OR, 2.778; 95% CI, 1.020–6.993 *p* = 0.031) irAEs than those harboring the *AA* genotype. Thus, the existence of the *PDCD1 PD-1.6* polymorphism (*G* allele) was associated with the occurrence of severe and multiple irAEs in patients with mRCC. Further evaluation of *PDCD1* polymorphisms might help identify patients experiencing irAE by nivolumab treatment.

## 1. Introduction

Renal cell carcinoma (RCC) is more common in males above 60 years of age [1], and 20–30% of patients with RCC already display the metastatic phenotype at the time of diagnosis. Survival rates for patients with metastatic RCC (mRCC) have remarkably improved since the introduction of tyrosine kinase inhibitors (TKIs) and immune checkpoint inhibitors (ICIs). Nivolumab, a novel anti-programmed cell death protein 1 (PD-1) antibody, has been applied in standard therapeutic regimens for patients with mRCC as second-line therapy and others [2]. Additionally, first-line nivolumab combination therapy with cytotoxic T-lymphocyte-associated protein 4 inhibitor or first-line cabozantinib therapy has been approved by the Food and Drug Administration, and these drugs are expected to contribute towards a better clinical outcome in patients with mRCC [3,4]. However, the immune-related adverse events (irAEs), which are undesirable effects of these therapies, remain unresolved. In three pivotal trials, CheckMate 025, CheckMate 214, and CheckMate 9ER, 19%, 46%, and 19.1% of patients experienced Grade 3 or higher irAE, and 8%, 22%, and 12.2% discontinued treatment, respectively [2,3,4]. Some irAEs can induce life-threatening events, such as myocarditis or myasthenia gravis [5,6,7].

Some studies have recently shown that the manifestation of an irAE and a durable therapeutic response might be related to each other [8,9]. In contrast, steroid treatment applied to cope with irAEs can reduce favorable antitumor effects [10]. Such a paradoxical situation has been frequently observed in clinical settings [11,12]. The manifestation of nivolumab toxicity can vary and is difficult to predict prior to treatment initiation [12,13]. As inhibition of PD-1 by nivolumab induces host immunity upregulation, the patient’s genetic background may play a crucial role in determining the effect of nivolumab [14].

Identifying biological markers for predicting the clinical effect and safety of therapeutic agents is highly desirable because of the increasing number of treatment options for advanced renal cancer. Although ICIs are available, a definitive strategy for treatment, including the appropriate choice or combination of agents with respect to patients’ health background, has not been developed. There are several approaches to identify clinically useful genetic variations. For instance, single nucleotide polymorphisms (SNPs) reportedly influence the effect of nivolumab [15]. A SNP is a genomic variant at a single base position in the DNA and may influence promoter activity (gene expression), messenger RNA (mRNA) conformation (stability), and subcellular localization of mRNAs and/or proteins and hence may affect the development of disease. Some studies have shown that SNPs in the PD-1 protein-coding gene, *PDCD1*, are related to a better survival rate [16] or worsened toxicity [17] in patients with malignancies treated by nivolumab.

Many SNPs in the *PDCD1* gene have been previously analyzed. The SNP *PD-1.3* located in intron 4 (rs11568821) was first described by Prokunina et al. to be associated with susceptibility to systemic lupus erythematosus [14]. An allele of SNP *PD-1.3* may alter a binding site of runt-related transcription factor 1 located in an intronic enhancer region and has been thought to increase susceptibility to autoimmunity [18]. Another SNP, *PD-1.5*, in exon 5 (rs2227981) has been shown to be associated with ankylosing spondylitis [19]. Other studies have further shown that PD-1 expression in CD4+ T cells is significantly lower in individuals with *PD-1.5 CC* genotype than in those with *PD-1.5 CT* and *PD-1.5 TT* genotypes [20]. *PD-1.6* is an SNP in the 3′-untranslated region (3′-UTR) of PD-1 (rs10204525) and is reportedly associated with susceptibility and disease progression in chronic hepatitis B virus infection [21]. The *PD-1.6* polymorphism reportedly alters cytokine production and PD-1 expression in peripheral blood mononuclear cells [22]. Nevertheless, a definitive overview of *PDCD1* SNP frequencies largely remains elusive owing to a large variation among different ethnic groups and a relative paucity of extensive studies [23]. In addition, no study has demonstrated the association between *PDCD1* SNPs and the clinical outcomes of patients with mRCC treated with nivolumab. Therefore, we aimed to investigate three common SNPs in *PDCD1* in Japanese patients with mRCC and their potential association with the clinical outcomes of patients treated with nivolumab. Our findings could form the basis for further evaluation of *PDCD1* polymorphisms, which might help identify patients benefitting from nivolumab treatment.

## 2. Materials and Methods

### 2.1. Patient Population

This retrospective study included 106 patients with mRCC treated with first-line, second-line, or other nivolumab regimens, between March 2013 and March 2019. Treatments were administered without any interruptions between cycles unless disease progression or intolerable toxicities were observed. Patients were classified according to the International Metastatic Renal Cell Carcinoma Database Consortium (IMDC) risk scoring system prior to nivolumab regimen initiation. The choice of first- or second-line systemic therapy was based on the IMDC risk score, performance status, disease extent, comorbidity presence, previous treatment, and individual preferences. Some patients underwent metastasectomies. The response to the nivolumab regimen was evaluated according to Response Evaluation Criteria in Solid Tumor (RECIST), version 1.1. The assessment interval for individual patients was scheduled by the attending physicians. All adverse events (AEs) were graded in concordance with the National Cancer Institute Common Toxicity Criteria (NCI-CTC), version 5.0. We obtained AE documentation from the patient medical records and investigated the relationship between SNPs and AEs of ≥ Grade 2. Patients who had more than three ≥ Grade 2 AEs were defined as having multiple AEs. All AEs leading to nivolumab treatment termination were noted.

### 2.2. Genetic Analysis

We analyzed three representative SNPs, namely *PD-1.3* (rs11568821), *PD-1.5* (rs2227981), and *PD-1.6* (rs10204525), in the *PDCD1* gene. The relationships between these SNPs and patient survival and clinical outcome, including progression-free survival (PFS), overall survival (OS), best objective response (BOR), and AEs, were evaluated. DNA was extracted from peripheral blood samples using a QIAamp Blood kit (Cat. No. ID 51104, Qiagen, Hilden, Germany) and stored at −20 °C until analysis. We followed its protocol handbook. Each SNP was genotyped using the PCR-restriction fragment length polymorphism method. Primer sequences, restriction enzymes, and PCR-restriction fragment length polymorphism product sizes are presented in Appendix A. *PD-1.6* was amplified as described previously by Direskeneli et al. using PCR [24], and the primers for *PD-1.3* and *PD-1.5* were designed using Primer 3Plus (https://www.bioinformatics.nl/cgi-bin/primer3plus/primer3plus.cgi: accessed on 1 November 2018). DNA was amplified in a Veriti^TM^ thermal cycler (Applied Biosystems, Waltham, MA, USA): starting with an initial denaturation at 95 °C for 10 min, followed by 35 cycles of denaturation at 95 °C for 30 s, annealing at 58 °C for 30 s, and elongation at 72 °C for 30 s, and, at last, a final elongation at 72 °C for 10 min.

### 2.3. Statistical Analyses

PFS was defined as the time between initiation of nivolumab treatment and disease progression or death, as confirmed by radiological images or clear clinical manifestation of progressive disease (PD). OS was defined as the time between nivolumab treatment initiation and death. The data records were closed upon patient death or final follow-up. The data are expressed as the median and range, and differences with a *p*-value < 0.05 were considered statistically significant. The chi-square test was used to examine differences in categorical data. PFS and OS data were stratified using the Kaplan–Meier method and analyzed using the log-rank test. The Cox proportional hazard regression model was used for the analysis of hazard ratio (HR) and 95% confidence interval (CI). To reduce the effects of selection biases and potential confounders in this observational study, we performed multiple logistic regression analyses. A multivariate analysis was performed to determine the odds ratios (ORs) and 95% CIs for factors with *p*-value < 0.1 in the univariate analysis. Statistical analyses were performed using SPSS statistical software, version 26.0 (SPSS Japan Inc., Tokyo, Japan). To test the population homogeneity among the subjects, the genotype frequencies of each polymorphism were tested against the Hardy–Weinberg equilibrium using the chi-square test.

### 2.4. Ethical Statement

All patients gave informed consent for this study, and the ethics committee of the Akita University Graduate School of Medicine and the Hirosaki Graduate School of Medicine approved the study (approved No. 1517 at Akita University Ethical Committee).

## 3. Results

### 3.1. Patient Characteristics

The patients’ clinical characteristics are described in Table 1. The median age was 69 years old (interquartile range [IQR]: 62–74 years old). All 106 patients were Japanese, and, among them, 85 were male (80%) and 21 were female (20%). Sixty-four (60%) patients had undergone radical nephrectomy prior to initiation of systemic therapies. Forty-seven (44%) patients were administered nivolumab and ipilimumab as first-line systemic therapy, whereas 59 were administered (56%) nivolumab as second-line therapy or at a later time point. Among the patients who received second-line or later therapies, 27 (26%) had previously been treated with one therapy and 15 (14%) with two or more therapies, and 17 (16%) patients showed no records of prior therapy. Based on the IMDC risk classification system, 15 patients (14%) were classified as favorable-risk, 47 (44%) as intermediate-risk, and 42 as poor-risk (40%). Twenty-nine (27%) patients had one metastatic site, 39 (37%) had two, and 38 (36%) had three or more. The sites of metastasis were the lungs (*n* = 79, 75%), bones (*n* = 36, 34%), lymph nodes (*n* = 30, 28%), adrenal glands (*n* = 15, 14%), liver (*n* = 14, 13%), brain (*n* = 10, 9%), and others (*n* = 27, 25%).

### 3.2. Patient Outcomes and Genotypic Characteristics

The median follow-up period after nivolumab initiation was 18.8 months (IQR, 6.3–33.9 months). At the time of data analysis, 26 (25%) patients continued with nivolumab therapy. Forty-five (42%) patients stopped nivolumab treatment owing to disease progression, whereas 14 (13%) stopped treatment due to the occurrence of AEs. Thirty-seven patients (35%) had died at the time of data analysis. The estimated median PFS and OS were 5.0 (IQR, 2.7–17.4) and 18.8 (IQR, 6.3–33.9) months, respectively. The BOR assessment data are shown for 103 patients (Table 1). Three patients who did not receive the first evaluation (early discontinuation) were excluded. Eight patients (8%) exhibited complete response (CR), 30 (28%) showed partial response (PR), 39 (37%) showed stable disease (SD), and 26 (24%) showed PD. Fifty-three (50%) of 106 patients developed AEs ≥ Grade 2, and 14 (13%) patients discontinued nivolumab because of AEs (Table 1).

SNPs were successfully analyzed in the 106 patients. None of the patients had the *PD-1.3* polymorphism. No significant relationship was observed between the genotypes of the two SNPs and PFS and OS (Table 2 and Figure 1). The distribution of SNPs in patients exhibiting CR + PR + SD vs. PD or in those displaying CR + PR vs. SD + PD was analyzed. No associations were detected between the clinical factors, SNPs, and the BOR (Table 3). Furthermore, there were no significant associations between clinical factors and polymorphisms and the risk of development of any AE (Table 4). In contrast, the incidence of ≥ Grade 2 irAEs was significantly higher in patients carrying the *PD-1.6 G* allele than in those carrying other genotypes (OR: 3.390, 95% CI: 1.517–7.576, *p* = 0.003; Table 4). In addition, patients harboring the *PD-1.6 G* allele had a significantly higher risk of multiple AEs than those harboring the *AA* genotype (OR: 2.778, 95% CI: 1.020–6.993, *p* = 0.031; Table 4). These instances of statistical significance were also confirmed by the logistic regression analysis performed to reduce selection biases and confounders (Appendix A). Although clinical stage 3 (OR: 3.469, 95% CI: 1.135–10.602, *p* = 0.029), prior history of nephrectomy (OR: 2.780, 95% CI: 1.120–6.896, *p* = 0.027) and difference of treatment regimen (OR: 3.879, 95% CI: 1.500–10.033, *p* = 0.005) were significant factors in univariate analysis (Table 4), these significances disappeared with the multi-valuable analysis (Appendix A).

## 4. Discussion

Presently, as the known candidate targets of nivolumab mainly originate from the tumor and the surrounding tumor microenvironment, programmed death-ligand 1 (PD-L1) expression in tumor cells measured by immunohistochemistry could predict responses to ICIs [25]. However, tumors that do not express detectable levels of PD-L1 on the cell surface can also respond to ICIs [3]. A recent study reported that PD-L1 expression is altered even during the course of clinical treatment [26]. This observation might lead to the notion that PD-L1 expression measured by immunohistochemistry cannot be reliable as a predictive biomarker of therapeutic response. Therefore, the development of better biomarkers for predicting responses to nivolumab therapy is necessary to help clinical decision-making and potentially expose fewer patients to inadequate treatments and the associated toxicities and costs [27,28].

In this study, we analyzed the potential of three PD-1 SNPs as a predictive biomarker of nivolumab therapeutic outcome in patients with mRCC, because PD-1 is the primary target of this anti-PD-1 antibody [29]. PD-1, which is broadly expressed in multiple cell types, including T cells, B cells, dendritic cells, and macrophages, plays a role in down-regulating T-cell responses, leading to immune suppression. Inhibition of PD-1 by nivolumab is a key aspect of the PD-1 pathway during signal induction [30]. The PD-1 protein is encoded by *PDCD1* and is expressed on CD4+ and CD8+ T cells. The functions of PD-1 polymorphisms have not yet been fully investigated; however, they may modify anti-tumor immunity and affect the response to ICIs [31].

As *PD-1.6 G > A* polymorphism is located at the 3′-UTR, it may affect transcriptional initiation, resulting in a decreased expression of PD-1 [32]. Although the *PD-1.6 A* allele is a minor allele among Caucasians (at most, 10%), the *A* allele is the major allele in Asian populations, appearing in 60–70% of the individuals [33]. The promotion of an immunosuppressive phenotype by *PD-1.6 G > A* polymorphism may also contribute to tumor development; nonetheless, this has not yet been characterized [31]. Zhang et al. reported that the *PD-1.6 A* allele increases PD-1 expression [21]. In the regulatory pathway, miRNA-4717 plays a crucial role [21]; miRNA-4717 has demonstrated allele-specific influence on luciferase activity in a dose-dependent manner in cells transfected with vectors containing different *PD-1.6 G* alleles. In patients with the *PD-1.6 GG* genotype, miRNA-4717 mimics a significantly decreased PD-1 expression [21]. In general, a high PD-1 expression seems to be related to a substantial antitumor effect and the development of irAEs induced by nivolumab administration [15]. However, the quantity of PD-1 protein is reduced in children with new-onset autoimmune diseases and type I diabetes compared with those negative for auto-antibody (AAB) presence [34]. Interestingly, the proportion of circulating FOXP3+ regulatory T cells expressing PD-1 has also been found to be reduced in children with an AAB-positive at-risk status compared with those in the control group showing an AAB-negative status [34]. This finding might support the argument that patients carrying the *PD-1.6 G* allele develop irAEs more frequently than those carrying the *AA* genotype. These significant associations suggest that individualized treatment strategies should be possible for patients carrying the *PD-1.6 G* allele. Although the *PD-1.5 T* allele showed poor survival outcomes in melanoma patients [16], no significant survival impact was observed in our study. Patients who carried a *PD-1.5 T* allele were significantly associated with lower mRNA expression of *PDCD1* in subcutaneous adipose tissue, not in lung nor liver as major metastatic organs by RCC [16]. This tissue specificity of mRNA expression might cause a different result in our study.

Nevertheless, this study has a few potential limitations. First, due to the retrospective study design, the treatment schedule, AE management, and timings for objective assessments could not be unified. Second, a validation study confirming the results of the SNP analysis was not conducted in the present study, which remains essential. Third, this study included patients treated with nivolumab monotherapy and ipilimumab and nivolumab combination therapy. The data from patients treated with the combination therapy might be affected by other factors, such as cytotoxic T-lymphocyte-associated protein 4-related genomic polymorphisms. Lastly, the major limitation is the lack of statistical power calculation, which might have led to a type II error because of the small sample size. However, to the best of our knowledge, this is the first study to report the clinical value of *PDCD1* SNPs in patients with mRCC who were treated with nivolumab. Further research is warranted to address the clinical implications of *PDCD1* SNPs in patients with mRCC.

## 5. Conclusions

Our retrospective assessment of SNPs in *PDCD1* as predictive markers of nivolumab toxicity in Japanese patients with mRCC suggested that the *PD-1.6 G* allele of *PDCD1* might be associated with increased severity and multiplicity of AEs. Upon further validation of our findings using a larger sample size, alternative mRCC treatment approaches based on *PDCD1* SNPs could be established in the future.

## Figures and Tables

**Figure 1 genes-13-01204-f001:**
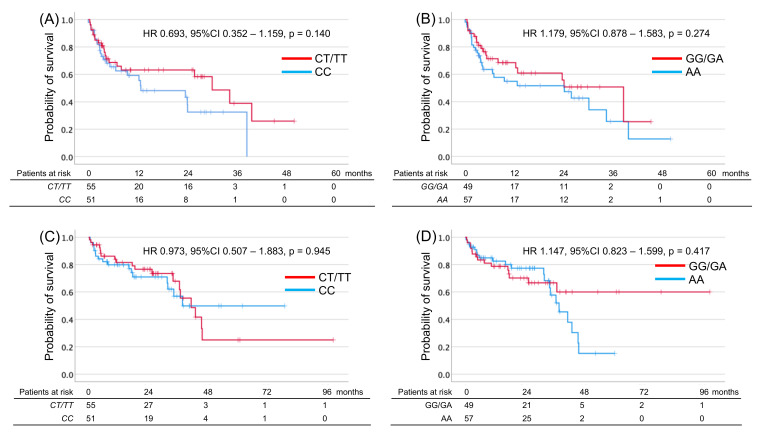
Kaplan–Meier estimates of progression-free survival (PFS) and overall survival (OS) in metastatic renal cell carcinoma patients and harboring *T* allele of *PDCD1* rs2227981 (*PD-1.5*) and *G* allele of *PDCD1* rs10204525 (*PD-1.6*). PFS (**A**,**B**) or OS (**C**,**D**) according to *PDCD1* genotype in *PD-1.5* (**A**,**C**) or *PD-1.6* (**B**,**D**).

**Table 1 genes-13-01204-t001:** Patients’ characteristics.

Number of Patients 106 (100%)			
Gender (%)		Number of metastatic sites	
Male	85 (80)	1	29 (27)
Female	21 (20)	2	39 (37)
Age (year)		≥3	38 (36)
Median	69	Site of metastasis and recurrence	
IQR	62–74	Lung	79 (75)
Histology (%)		Primary site	37 (35)
Clear cell	100 (94)	Bone	36 (34)
Others	6 (6)	Lymph node	30 (28)
Prior nephrectomy (%)		Adrenal	15 (14)
Yes	64 (60)	Liver	14 (13)
Agent (%)		Brain	10 (9)
Nivolumab	59 (56)	Contralateral kidney	6 (6)
Ipilimumab + Nivolumab	47 (44)	Others	27 (25)
Observational period (months)		IMDC risk classification	
Median	18.8	Favorable	15 (14)
IQR	6.3–33.9	Intermediate	47 (44)
Treatment duration (months)		Poor	42 (40)
Median	5.0	Unclassified	2 (2)
Range	2.7–17.4	Best response	
Number of prior systemic therapy (%)		CR	8 (8)
0	47 (44)	PR	30 (28)
1	27 (26)	SD	39 (37)
≥2	15 (14)	PD	26 (24)
unknown	17 (16)	Unknown	3 (3)
Clinical stage at diagnosis of RCC (%)		Reason for nivolumab discontinuation	
1	14 (13)	PD	45 (36)
2	6 (6)	AE	14 (13)
3	15 (14)	Still continue	26 (26)
4	55 (52)	Suspension	5 (24)
unknown	16 (15)	Other reasons	16 (15)
		irAE Grade 2 or higher	
			53 (50)
		Multiple irAEs	
			26 (25)

IQR, interquartile range; RCC, renal cell carcinoma; IMDC, International Metastatic Renal Cell Carcinoma Database Consortium; CR, complete response; PR, partial response; SD, stable disease; PD, progressive disease; AE, adverse event; irAE, immune-related adverse event.

**Table 2 genes-13-01204-t002:** Association between survival outcome and *PDCD1* polymorphism.

Gene	RS	Position	Genotype	Number of Patients	Risk Genotype	Progression-Free Survival (Months)			Overall Survival (Months)		
						HR	95% CI	*p*	HR	95% CI	*p*
*PDCD1*	rs2227981	*PD-1.5*	*CC*	51	*T allele*	0.639	0.352–1.159	0.140	0.973	0.507–1.883	0.945
		*C > T*	*CT*	53							
			*TT*	2							
	rs10204525	*PD-1.6*	*GG*	11	*G allele*	1.179	0.878–1.583	0.274	1.147	0.823–1.599	0.417
		*G > A*	*GA*	38							
			AA	57							

*PDCD1*, *Programmed cell death protein 1*; RS, reference single nucleotide polymorphism identification number; HR, hazard ratio; CI, confidence interval.

**Table 3 genes-13-01204-t003:** Distributions of clinical factors and *PDCD1* polymorphisms for the response.

Factor	Risk Category	Clinical Benefit			ORR		
		OR	95% CI	*p*	OR	95% CI	*p*
Age	69≦	1.330	0.544–3.252	0.532	0.706	0.316–1.575	0.395
Sex	Male	1.508	0.457–4.971	0.500	0.650	0.229–1.847	0.419
Nephrectomy	Yes	1.033	0.416–2.567	0.944	1.450	0.644–3.265	0.369
Regimen	Nivo + Ipi	0.571	0.227–1.434	0.233	0.608	0.272–1.355	0.233
Clinical stage	3≦	1.520	0.450–5.130	0.500	1.440	0.525–3948	0.479
	4	1.684	0.611–4.640	0.313	0.929	0.386–2.232	0.868
IMDC	Poor	1.655	0.675–4.056	0.271	1.297	0.570–2.951	0.379
	Intermediate + Poor	5.556	0.693–44.513	0.106	0.578	0.170–1.960	0.695
Number of Metastatic Organ	2≦	3.761	1.034–13.682	0.044	0.902	0.368–2.211	0.822
	3≦	2.841	1.142–7.065	0.025	1.290	0.554–3.001	0.555
irAE	G2≦	0.551	0.223–1.361	0.196	0.498	0.221–1.119	0.091
	G3≦	0.766	0.296–1.982	0.583	0.750	0.328–1.713	0.494
	multiple	1.931	0.732–5.093	0.184	1.096	0.140–4.129	0.750
*PDCD1* SNP	*PD-1.5 T* allele	0.755	0.310–1.836	0.536	0.562	0.250–1.261	0.162
	*PD-1.6 G* allele	0.791	0.323–1.935	0.608	1.335	0.598–2.978	0.481

IMDC, International Metastatic Renal Cell Carcinoma Database Consortium; *PDCD1*, *Programmed cell death protein 1*; Nivo + Ipi, nivolumab plus ipilimumab; PD, progressive disease; G, grade; ORR, objective response rate; OR, odds ratio; CI, confidence interval.

**Table 4 genes-13-01204-t004:** Association between adverse events and clinical factors and polymorphisms in *PDCD1* gene.

Factor	Risk Category	At Least One irAE ≥ G2			Multiple irAEs		
		OR	95% CI	*p*	OR	95% CI	*p*
Age	69≦	0.764	0.354–1.646	0.491	1.135	0.468–2.755	0.779
Sex	Male	0.909	0.349–2.368	0.845	0.442	0.119–1.643	0.223
Nephrectomy	Yes	0.952	0.435–2.986	0.903	2.780	1.120–6.896	0.027
Regimen	Nivo + Ipi	0.561	0.258–1.221	0.145	3.879	1.500–10.033	0.005
Clinical stage	3≦	3.469	1.135–10.602	0.029	0.817	0.262–2.548	0.727
	4	0.824	0.375–1.810	0.629	1.040	0.404–2.675	0.935
IMDC	Poor	0.824	0.375–1.810	0.629	1.138	0.458–2.829	0.781
	Intermediate + Poor	0.457	0.144–1.445	0.187	0.180	0.022–1.442	0.106
Number of Metastatic Organ	2≦	1.160	0.698–3.944	0.251	0.730	0.260–2.953	0.551
	3≦	0.893	0.401–1.990	0.782	0.540	0.218–1.335	0.182
*PDCD1* SNP	*PD-1.5 T* allele	0.708	0.328–1.527	0.379	1.324	0.545–3.221	0.536
	*PD-1.6 G* allele	3.390	1.517–7.576	0.003	2.778	1.020–6.993	0.031

IMDC, International Metastatic Renal Cell Carcinoma Database Consortium; *PDCD1*, *Programmed cell death protein 1*; SNP, single nucleotide polymorphism; Nivo + Ipi, nivolumab plus ipilimumab; AE, adverse event; irAE, immune-related adverse event; G, grade; OR, odds ratio; CI, confidence interval.

## Data Availability

The data presented in this study are available on request from the corresponding author.

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
