# Peer review of "Severe Immune-Related Adverse Events in Patients Treated with Nivolumab for Metastatic Renal Cell Carcinoma Are Associated with PDCD1 Polymorphism"

_genes, 2022, doi:10.3390/genes13071204_

Round 1
Reviewer 1 Report
I have not been able to see the authors' cover letter regarding this resubmission but none of the concerns in the 1st review have been addressed, other than increasing the number of patients treated with ipi/nivo.
Author Response
Please find the points to points responses in the attachment. Thank you.

Reviewer 2 Report
Mizuki Kobayashi and associates in their paper report the results of study devoted to analysis of the relationship between the response of patients with metastatic renal carcinoma (mRCC) to nivolumab based therapy and single nucleotide polymorphism (SNPs) of programed cell death protein 1 (PD-1) coding gene (PDCD1). The authors finding points to the possible relation between clinical outcomes of patients with mRCC and kind of SNP in PDCD1. Authors analyses indicate that patients with PD-1.6 G allele more frequently experienced immune-related adverse events in response to nivolumab treatment.
Overall, the study was properly designed, although authors indicate several potential limitations of their work. Moreover, owing to small sample size the conclusions are weakly supported statistically. Nevertheless, the reported results are interesting and provide data which upon validation on larger sample size could provide useful genetic markers that would help design effective and safe treatment strategy for patients with mRCC.
Author Response
We wish to express our appreciation to reviewer #2 for their insightful comments on our paper. The comments have helped us significantly improve the paper.
Mizuki Kobayashi and associates in their paper report the results of study devoted to analysis of the relationship between the response of patients with metastatic renal carcinoma (mRCC) to nivolumab based therapy and single nucleotide polymorphism (SNPs) of programed cell death protein 1 (PD-1) coding gene (PDCD1). The authors finding points to the possible relation between clinical outcomes of patients with mRCC and kind of SNP in PDCD1. Authors analyses indicate that patients with PD-1.6 G allele more frequently experienced immune-related adverse events in response to nivolumab treatment.
Overall, the study was properly designed, although authors indicate several potential limitations of their work. Moreover, owing to small sample size the conclusions are weakly supported statistically. Nevertheless, the reported results are interesting and provide data which upon validation on larger sample size could provide useful genetic markers that would help design effective and safe treatment strategy for patients with mRCC.
Response: We thank the reviewer’s comment. Your comment convinced us to conduct a larger-scale study to validate the potential of PD-1.6 polymorphism as a predictive marker in mRCC patients. Thank you again for your optimal comment.
Reviewer 3 Report
The manuscript titled “Severe immune-related adverse events in patients treated with nivolumab for metastatic renal cell carcinoma are associated with PDCD1 polymorphism” describes the Japanese metastatic renal cell carcinoma patients with PDCD1 PD-1.6 polymorphism showed severe and multiple immune-related adverse effects treated by nivolumab. The followings are some concerns and comments have been pointed out that the authors may want to consider.
1. Line 19: Please use italic p as it refers to a p-value. Check throughout the manuscript.
2. Line 26 Introduction section: I’d suggest the authors reorganize this part by splitting it into 3 or 4 paragraphs to make it easier to track and read.
3. Line 100: Please include CAT# for the kit. Check throughout the manuscript.
4. Line 101: At least a brief DNA extraction protocol should include.
5. Line 104 Table S1: Please provide PCR gel image results instead of only a supplementary table. And please perform additional analysis if possible.
6. Line 116 and line 123: Please homogenous the format throughout the manuscript with or without a space before and after signs, for example, “<”.
7. Lines 118-119: Please provide PFS and OS survival curve figures. I’d suggest the authors include them in the main context even though the results are negative.
8. Line 131: I’d suggest the authors use “years old” for age instead of “years”. Check throughout the manuscript.
9. Line 140: The “29 (37%) had two” should be “27%”.
10. Line 147: A space is needed between “18.8” and “months”.
11. Lines 147-157: Please homogenous the format throughout the manuscript by adding percentages followed by the numbers of patients for easier tracking and reading.
12. Line 171 Table 2: a) I’d suggest the authors just use “PFS”, “OS”, and “n” (n = number of patients) in the table to save some space and look better. b) I’d suggest the authors add percentage followed the number n. c) The value “0.1.40” ( should be “0.140”, “2,652” should be “2.652”
13. Line 173 Table 4: The second last value of the third column should be “0.708” instead of “0708”.
14. Line 173 Table 4: I’d highly suggest the authors add more description and discussion for p=0.029 (column 5 line 5), p=0.027 (last column line 3), and p=0.005 (last column line 4).
15. Line 176 Discussion section: I’d suggest the authors reorganize this part to make it more logical and smooth. For example, the first paragraph seems not suitable as the first paragraph in the discussion part.
Author Response
We wish to express our appreciation to reviewer #3 for their insightful comments on our paper. The comments have helped us significantly improve the paper.
The manuscript titled “Severe immune-related adverse events in patients treated with nivolumab for metastatic renal cell carcinoma are associated with PDCD1 polymorphism” describes the Japanese metastatic renal cell carcinoma patients with PDCD1 PD-1.6 polymorphism showed severe and multiple immune-related adverse effects treated by nivolumab. The followings are some concerns and comments have been pointed out that the authors may want to consider.
Response: We thank the reviewer’s comment. All papers suggested by the reviewer were useful and could promote the reliability of our results.
- Line 19: Please use italic p as it refers to a p-value. Check throughout the manuscript.
Response: We thank the reviewer’s comment. All papers suggested by the reviewer were useful and could promote the reliability of our results. In accordance with the reviewer’s comment, we checked all “p” and revised if any mistakes existed.
- Line 26 Introduction section: I’d suggest the authors reorganize this part by splitting it into 3 or 4 paragraphs to make it easier to track and read.
Response: Thank you for your valuable comment. We are sorry for the inconvenience to the reviewer by the too-long paragraph in the introduction. We divided the introduction into 4 paragraphs for the purpose of better readability.
- Line 100: Please include CAT# for the kit. Check throughout the manuscript.
Response: Thank you for your query to improve our paper. We inserted the CAT# to line 101 as follows:
Cat. No. ID 51104
- Line 101: At least a brief DNA extraction protocol should include.
Response: Thank you for your query to improve our paper. We are sorry for the inconvenience to the reviewer by racking the DNA extract protocol; however, the protocol is a little bit complicated to put in the material section and available easily via the QIAGEN website. So, we inserted a sentence into line 102 as follows:
We followed its protocol handbook.
- Line 104 Table S1: Please provide PCR gel image results instead of only a supplementary table. And please perform additional analysis if possible.
Response: Thank you for your query to improve our paper. We are wondering if the reviewer wanted to know the real results of PCR-RFLP. We showed partial results of the image of the RFLP-PCR to PD-1.6 only for the reviewer as follows:
- Line 116 and line 123: Please homogenous the format throughout the manuscript with or without a space before and after signs, for example, “<”.
Response: We thank the reviewer’s comment. In accordance with the reviewer’s comment, we checked the space before and after signs.
- Lines 118-119: Please provide PFS and OS survival curve figures. I’d suggest the authors include them in the main context even though the results are negative.
Response: Thank you for your valuable comment. We totally agree with your suggestion and added a new figure of PFS and OS as Figure 1 in line 177.
- Line 131: I’d suggest the authors use “years old” for age instead of “years”. Check throughout the manuscript.
Response: We thank the reviewer’s comment. In accordance with the reviewer’s recommendation, we revised “years” to “years old” in line 136.
- Line 140: The “29 (37%) had two” should be “27%”.
Response: Thank you for your query to improve our paper. We are sorry for the inconvenience to the reviewer by our mistake. We corrected the mistake “29 (37%)” to 39 (37%)
- Line 147: A space is needed between “18.8” and “months”.
Response: We thank the reviewer’s comment. In accordance with the reviewer’s comment, we inserted a space between “18.8” and “months”.
- Lines 147-157: Please homogenous the format throughout the manuscript by adding percentages followed by the numbers of patients for easier tracking and reading.
Response: We thank the reviewer’s comment. In accordance with the reviewer’s comment, we inserted the percentages as follows:
At the time of data analysis, 26 (25%) patients continued with nivolumab therapy. For-ty-five (42%) patients stopped nivolumab treatment owing to disease progression, whereas 14 (13%) stopped treatment due to the occurrence of AEs. Thirty-seven patients (35%) died at the time of data analysis.
- Line 171 Table 2: a) I’d suggest the authors just use “PFS”, “OS”, and “n” (n = number of patients) in the table to save some space and look better. b) I’d suggest the authors add percentage followed the number n. c) The value “0.1.40” ( should be “0.140”, “2,652” should be “2.652”
Response: Thank you for your valuable comment. We are sorry for the inconvenience to the reviewer by our mistakes and found some issues in Table 2. We corrected all the mistakes in Table 2. Could you kindly check them in the manuscript?
- Line 173 Table 4: The second last value of the third column should be “0.708” instead of “0708”.
Response: Thank you for your careful and kind checking. In accordance with the reviewer’s comment, we corrected “0708” to “0.708”. Thank you again.
- Line 173 Table 4: I’d highly suggest the authors add more description and discussion for p=0.029 (column 5 line 5), p=0.027 (last column line 3), and p=0.005 (last column line 4).
Response: Thank you for your critical comment. We are sorry for the inconvenience to the reviewer by racking these important findings. We agree with the reviewer’s comment and inserted a sentence of the results of the univariate analysis into line 176 as follows:
Although clinical stage 3 (OR: 3.469, 95% CI: 1.135– 10.602, p = 0.029), prior history of nephrectomy (OR: 2.780, 95% CI: 1.120– 6.896, p = 0.027) and difference of treatment regimen (OR: 3.879, 95% CI: 1.500– 10.033, p = 0.005) were significant factors in univariate analysis (Table 4), these significance were disappeared by the multi-valuable analysis (Table S2)
- Line 176 Discussion section: I’d suggest the authors reorganize this part to make it more logical and smooth. For example, the first paragraph seems not suitable as the first paragraph in the discussion part.
Response: Thank you for your valuable comment. In accordance with the reviewer’s comment, we deleted the first paragraph

Round 2
Reviewer 3 Report
I do not have any further concerns now. Thank you. Please make sure the images in Figure 1 are clear (they are not clear in the current version) before publication. Good luck.
Author Response
We wish to thank reviewer #1 for these comments on our paper. We feel the comments have helped us significantly improve the paper.
This manuscript is a resubmission of an earlier submission. The following is a list of the peer review reports and author responses from that submission.
Round 1
Reviewer 1 Report
- This is a mix of patients on first line and second line treatment, and nivo monotherapy and combination therapy. I suggest homogeneous inclusion criteria.
- PDCD1 includes multiple SNPs. Please provide a rationale why the three SNPs were chosen for the present study, given the fact that there is no data on these SNPs in RCC. Why were no other SNPs from other genes chosen?
- There is no adjustment for multiple testing, which should be done in view of the fact that multiple hypotheses were tested.
- Dichotomisation of continuous variables should be avoided.
Reviewer 2 Report
Summary The authors present a well-written manuscript regarding the possible use of SNPs in the PDCD1 gene as a biomarker for IO related toxicity from Nivolumab treatment in RCC in a Japanese population. The authors should be commended on trying to develop a biomarker for immunotherapy efficacy/toxicity in RCC where the treatment paradigm is currently so complex. Whilst there are a number of limitations to the study (mostly noted by the authors related to the frequency of the polymorphisms of interest, small sample size and a range of treatments being given including ipilimumab) and some additional clarity is required, the study adds new information to the field and may generate new hypotheses regarding multiple SNPs in mRCC patients in this and different populations. Minor Comments 1. In the simple summary the sentence "Patients harbouring the PD-1.6G allele..." does not read well with the word multiple. If accurate, would it be possible to say"...more severe and a higher occurrence of multiple IRAEs". 2. There is no evidence to support the last line of the abstract in this study- would be better to focus on IRAEs than discuss patients "benefiting" from Nivolumab 3. Introduction- line "However, the immune related adverse event...." are not unexpected. They are unwanted, but completely expected. 4. Introduction- In discussion about the CheckMate studies it would be helpful to include the % patients experiencing AEs and discontinuing treatment. 5. Introduction- there is a new melanoma paper looking at PDCD1 polymorphism which could also be referenced published in March 2021: "Germline Variation in PDCD1 Is Associated with Overall Survival in Patients with Metastatic Melanoma Treated with Anti-PD-1 Monotherapy". 6. Results- Should read "eighteen patients had died" rather than died. 7. Table 2: why is SNP PD-1.3 not included anywhere? Text talks about 3 SNPs and only 2 in the table. 8. Table 2: It would be helpful to explain what is meant by Risk genotype (e.g. ? CT and TT vs. CC) and to include WT genotype or a notation (e.g. G>A) to highlight this. 9. Authors correctly identify the study limitations and appropriately highlight these. 10. It may be helpful to include a line or two in the introduction regarding SNPs, Risk/minor alleles etc. to widen the target audience of the article. Major Comment 1. It is not clear throughout the manuscript when discussing the PD-1.6 G allele and the frequency/severity of IRAEs exactly what the comparison is between. i.e. Is it that patients with the PD1.6 G (GG/GA) allele have more severe and a higher number of IRAEs than patients with the PD 1.6 A (AA) allele or that patients with the PD1.6 G allele have more severe and a higher number of IRAEs than those with the PD1.3/PD 1.5 risk alleles (which is what is implied by the simple summary and abstract)? The former is much more interesting than the latter.